# Video abstracts and plain language summaries are more effective than graphical abstracts and published abstracts

**Kate Bredbenner**\* , **Sanford M. Simon**

Lab of Cellular Biophysics, Rockefeller University, New York, New York, United States of America

\* kate.bredbenner@gmail.com

## Abstract

### Background

Journals are trying to make their papers more accessible by creating a variety of research summaries including graphical abstracts, video abstracts, and plain language summaries. It is unknown if individuals with science, science-related, or non-science careers prefer different summaries, which approach is most effective, or even what criteria should be used for judging which approach is most effective. A survey was created to address this gap in our knowledge. Two papers from Nature on similar research topics were chosen, and different kinds of research summaries were created for each one. Questions to measure comprehension of the research, as well as self-evaluation of enjoyment of the summary, perceived understanding after viewing the summary, and the desire for more updates of that summary type were asked to determine the relative merits of each of the summaries.

### Results

Participants (n = 538) were randomly assigned to one of the summary types. The response of adults with science, science-related, and non-science careers were slightly different, but they show similar trends. All groups performed well on a post-summary test, but participants reported higher perceived understanding when presented with a video or plain language summary (p<0.0025). All groups enjoyed video abstracts the most followed by plain language summaries, and then graphical abstracts and published abstracts. The reported preference for different summary types was generally not correlated to the comprehension of the summaries. Here we show that original abstracts and graphical abstracts are not as successful as video abstracts and plain language summaries at producing comprehension, a feeling of understanding, and enjoyment. Our results indicate the value of relaxing the word counts in the abstract to allow for more plain language or including a plain language summary section along with the abstract.

**Data Availability Statement:** All relevant data are within the manuscript and its Supporting Information files.

**Funding:** We are grateful for the support of NIH 5R01GM119585 to SMS (https://www.nih.gov/). The funders had no role in study design, data collection and analysis, decision to publish, or preparation of the manuscript.

**Competing interests:** The authors have read the journal's policy and the authors of this manuscript have the following competing interests: KB is a paid creator of video abstracts via SimpleBiologist. This does not alter the authors' adherence to PLOS ONE policies on sharing data and materials.

## Introduction

Every scientific paper is a story, but it can be a challenge to access those stories. Many papers are hidden behind subscription fees that make access prohibitive. But even if the reader gets behind the paywall, scientific stories are often written in a dense and jargon-laden fashion. This kind of style may not be limiting for experts in the field, but for those outside of that field, it can ensure that story is not heard. This has led to a recent incorporation of different kinds of summaries with the goal of making the science more accessible.

In a recent 3M survey of 14,025 people, 88% of them thought that scientists should be sharing their results in easy to understand language [1]. Many journals have recognized this need, and they create a variety of summaries including videos, graphics, and plain language summaries in addition to the abstracts that come with every scientific paper. While all of these summaries tell the same story, they tell it using different media styles.

eLife is committed to plain language summaries and has been creating them since the journal was founded in 2012 [2]. Other journals have followed eLife's lead in incorporating plain language summaries including the Public Library of Science (PLOS) journal family, Proceedings of the National Academy of Sciences, USA (PNAS), Cell, and Science [3]. All Cochrane systematic reviews also require plain language summaries [4]. Despite these journals all writing plain language summaries, there is no recognized standard summary format. Plain language summaries are usually called different names and have different word counts depending on the journal. The one thing that remains constant is a lack of jargon.

Plain language summaries aren't the only summary type that has been introduced to reach a wider audience. Videos have become very popular in the last few years. Video abstracts are generally three to five minutes in length and cover the major findings of the research paper they are about. The first video abstract was produced by Cell in 2009 [5], and since then Cell has been a major contributor of video summaries. Other video contributors include the Wiley publishing group and ACS Publications.

Cell has also been a leader for graphical abstracts. Cell defines the graphical abstract as "one single-panel image that is designed to give readers an immediate understanding of the take-home message of the paper." They also say that "its intent is to encourage browsing, promote interdisciplinary scholarship, and help readers quickly identify which papers are most relevant to their research interests [6]."

While having different ways to summarize published research could increase accessibility, each of these summaries takes time to make. Video abstracts can take over 20 hours to complete and graphical abstracts aren't far behind. They also require specialized equipment and skills to be effective [7,8]. While plain language summaries might seem the easiest to produce, even eLife found that they were publishing too many papers for each one to have its own plain language summary, and in 2016 they scaled back the number of summaries they publish [9].

Summaries are necessary for sharing scientific findings quickly with peers and the public. Unfortunately, only a small portion of journals create even one kind of summary for their papers. To encourage more journals to summarize the research they publish, it would help to know what the most effective summaries are for different audiences.

Previous research on blogs which combined a written article with videos or graphics concluded that scientists had better recall and enjoyed the blog post more when a video was combined with the text whereas non-scientists had better recall and enjoyed the blog post more with an image included [10]. Research on support for the James Webb Space Telescope found that participants were more supportive of telescope construction when they viewed interactive media including a video and a simulation over traditional text [11]. We also know that people tend to be able to recognize science images better than they can answer science questions [12].

All of this data taken together suggests that videos or graphics might be more important when it comes to enjoyment and comprehension of science summaries, but we are still missing data which directly compares science summaries in the way that they are currently created by journals. Results of an eLife survey in 2016 found that they have a ratio of scientist: non-scientist readership of 6:1 for their plain language summaries, and over 90% of both scientists and non-scientists said that most or all of the summaries they've read were informative [13]. However, we don't know the relative efficacy of reaching people with different kinds of summaries. We also don't know if adults with science, science-related, and non-science careers all enjoy and comprehend the same kinds of summaries.

To evaluate the effectiveness of different summary types for people with different careers, we created a survey that presents participants with a video abstract, graphical abstract, plain language summary, or published abstract from two papers in the same subject area (S6 File). The survey looked at comprehension, perceived understanding, enjoyment, and whether the participants wanted to see more summaries of that type. The combination of these four measurements was used to determine which summary method is most effective. We also compared across career types and reported learning preferences to see what role they play. Finally, we offer suggestions to researchers and journals regarding what to do about summaries in the future.

## Methods

### Science summary design

This work was granted exempt status from the Rockefeller University IRB (ref #342107). We chose two recently published papers as the subject of study. Cohn et al. was published in Nature Medicine in April 2018 and outlines a method for recovering latent cells from HIV-1+ patient blood in order to study the latent cells for a possible future cure. It also sequences these latent cells and shows that they are often clonal [14]. Takata et al. was published in Nature in September 2017 and shows that HIV-1 has selectively removed CG dinucleotides from its genome to more closely mimic the nucleotide content of its human host. Specifically, HIV-1 has removed CG dinucleotides to avoid the host protein ZAP which recognizes and destroys RNA in the cell that has these dinucleotides [15]. Both papers are within the HIV-1 field, and both were published at similar times in similar journals. Both papers were also first-authored by graduate students both at Rockefeller University and both in the same year of graduate school.

The abstracts of both papers were taken, as published, to place into a survey. From there, a plain language summary was written for each paper (S5 File). The main takeaways that were highlighted in the published abstracts were also the main focus of the plain language summaries. Every effort was taken to eliminate any jargon and to provide real-world context for each of the findings. Both plain language summaries were of similar length (422 words for Cohn et al.; 433 words for Takata et al.). The plain language summaries followed the guidelines put into place by eLife, including their list of questions that they ask the scientists to make the summaries easier for the editors to write [16]. The abstracts and plain language summaries were put into a readability calculator to obtain the Flesch Reading Ease Score (FRES) and the Flesch-Kincaid Grade Level (FKGL) similar to previously published work on plain language summaries [17]. The scores can be seen in Table 1.

The plain language summaries were used as the spoken script for the video abstracts (S2 and S3 Files). The videos of each paper were illustrated using similar visual motifs and the videos were of similar lengths (2:33 for Cohn et al.; 2:49 for Takata et al.). Both videos followed the "whiteboard explainer" style where images are drawn on a screen and the drawing is either

**Table 1. Readability of published abstracts and plain language summaries for Cohn et al. and Takata et al.**

| | Cohn et al. | | Takata et al. | |
|---|---|---|---|---|
| | **Abstract** | **Plain Language** | **Abstract** | **Plain Language** |
| **FRES**[a] | 24.2 | 63 | 15.2 | 58 |
| **FKGL**[b] | 16.2 | 9.5 | 16.1 | 10 |

All scores were obtained from www.readability-score.com.

[a] Flesch Reading Ease Score

[b] Flesch-Kincaid Grade Level

freeze-framed or sped up to match the narration. Both videos were uploaded to YouTube and were made unlisted so only survey participants could see them. The YouTube generated closed captions were edited to reflect the actual script, and closed captions were set to automatically appear anytime the videos were played. The closed captions could be turned off by the participants if they wanted.

The videos followed all of the qualifications set by Cell for their video abstracts [18]. All technical specifications were met or exceeded, and the videos were within the requested length. The first author has made several of these Cell video abstracts and is familiar with the qualifications necessary. Audio quality was also carefully controlled as it plays a role in how favorable participants find the research [8].

The graphical abstracts were created using Keynote software and they followed the same visual motifs that were in the videos (S4 File). The graphical abstracts were placed into a color-blindness simulator to be sure that all possible participants could see the image equally well [19]. The graphical abstracts were also created based on the guidelines set up by Cell for their graphical abstracts [6]. All technical specifications were met or exceeded where possible.

All summaries were created with the intent that the videos, graphics, and plain language summaries should be content-identical.

## Survey design

The survey was created in the Google Forms platform. Eight surveys were created that were mostly identical except for the type of summary shown. Two surveys showed video abstracts, two showed graphical abstracts, two showed plain language summaries, and two showed published abstracts. Each pair of surveys showed both the Cohn et al. summary and the Takata et al. summary, but one version showed the Cohn et al. summary first and one showed the Takata et al. summary first (Fig 1). Switching which paper was shown first was done to be able to prove that the responses received did not differ based on which paper summary was seen first. Participants were randomly assorted to one of the eight surveys via a random URL generator embedded into the button on the survey website. Each participant only completed one survey meaning that they only saw one type of science summary, but they saw that type of summary for both papers (Fig 1).

Four of the surveys contained a collection error that was corrected part way through data collection (Fig 1). If participants were funneled to one of the surveys with the error, only participants who marked that they had science careers were shown both the Cohn et al. and Takata et al. summaries. All other participants only saw the Cohn et al. summary. This error lead to fewer non-science and science-related participants for the Takata et al. paper, but the error was corrected in time to still get usable data.

Before presenting the created summaries, all surveys asked participants to report their career type (science, science-related, non-science, or undergraduate), input their gender if

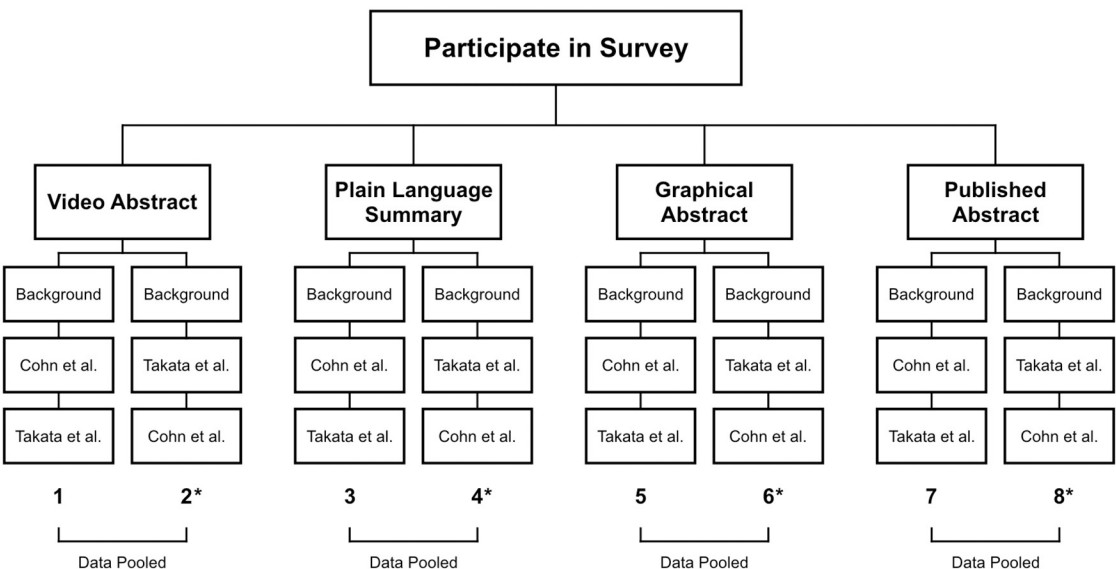

**Fig 1. Flowchart of survey assignment and pooling.** A flowchart representing the eight survey versions created for this research. Once participants click on the button "Participate in Survey", they are randomly assorted to one of the eight possible surveys including two versions each of video abstracts, plain language summaries, graphical abstracts, and published abstracts where one version shows the Cohn et al. summary first and one shows the Takata et al. summary first. All surveys ask background questions prior to showing a summary. The asterisks denote surveys that contained an error which only showed participants with science careers both the Takata et al. and Cohn et al. summaries. All other participants were only shown the Cohn et al. summary. The error was correctly shortly after publicizing the survey. Data from both versions of each type of summary were pooled, as denoted by brackets and the phrase "Data Pooled".

they so desired (a fill-in-the-blank that was not required), and report their preference for receiving science updates (written summaries, video, audio, reading the original research paper, or graphics) (S6 File).

Participants that reported they had science careers were asked an additional series of questions about how they prefer to receive research updates in their field versus outside of their field. This list of options included newspaper articles, social media, recommendations from friends and colleagues, scientific journals, and PubMed/other alerts (S6 File).

After the background information was collected, participants were shown one of the summaries and asked follow up questions about it. There was a 6-question quiz associated with each of the papers to determine comprehension (Table 2). The quiz was one multiple choice question and five true/false questions. These questions were designed to be answerable regardless of which summary type the participant had seen. Other follow up questions were asked to determine how much the participants enjoyed and understood the research and also how much they wanted to see more summaries of the type that they were presented (Table 2). The full survey is available as supplementary file 6.

## Survey recruitment

Recruitment was done using the snowball method used in studies similar to this research [10]. Participants were recruited online via the first author's social media pages using appropriate hashtags. Emails were also sent to a number of science groups including the National Alliance for Broader Impacts (NABI), The Falling Walls organization, the BioBus, all attendees of the 2019 SciOut conference, all attendees of the Science Alliance Leadership Training (SALT) up to 2018, and all members of the Rockefeller University Community.

**Table 2. Follow up questions for Cohn et al. and Takata et al.**

| Type of Question | Cohn et al. | Takata et al. |
|---|---|---|
| **Comprehension–Multiple Choice** | This research focuses on: | This research focuses on: |
| | **(a) HIV** | **(a) HIV** |
| | (b) FIV | (b) FIV |
| | (c) Influenza | (c) Influenza |
| | (d) I don't know | (d) I don't know |
| **Comprehension–T/F** | This research created a capture technique to collect all T-cells from patients. **False** | Vertebrates have evolved less AG nucleotide pairs. **False** |
| **Comprehension–T/F** | The capture technique is a type of cure for the virus discussed in the summary. **False** | The virus mentioned has evolved to lack CG pairs to avoid cell anti-viral defenses. **True** |
| **Comprehension–T/F** | Latent cells captured from patient blood are mostly from a single latent cell that divided. **True** | ZAP interacts with the DNA of the virus mentioned in the summary. **False** |
| **Comprehension–T/F** | Captured latent cells have higher expression of genes that increase virus activation. **False** | All possible DNA nucleotide pairs show up at the same rate as each other in vertebrates (eg. AT is present at the same frequency as GT or CG or GC). **False** |
| **Comprehension–T/F** | Latent cells are a consequence of the lifecycle of the virus mentioned. **True** | ZAP is a protein that is made by the infected host cell. **True** |
| **Enjoyment** | I enjoyed reading[a] this abstract[b]: | I enjoyed reading[a] this abstract[b]: |
| | (0) Not at all | (0) Not at all |
| | (1) A bit | (1) A bit |
| | (2) Average | (2) Average |
| | (3) Mostly | (3) Mostly |
| | (4) Very Much | (4) Very Much |
| **Understanding** | I understand this research more after reading[a] this abstract[b]: | I understand this research more after reading[a] this abstract[b]: |
| | (0) Not at all | (0) Not at all |
| | (1) A bit | (1) A bit |
| | (2) Average | (2) Average |
| | (3) Mostly | (3) Mostly |
| | (4) Very Much | (4) Very Much |
| **Desire for Updates** | I want to get more science updates via written abstract[b] after reading[a] this: | I want to get more science updates via written abstract[b] after reading[a] this: |
| | (0) Not at all | (0) Not at all |
| | (1) A bit | (1) A bit |
| | (2) Average | (2) Average |
| | (3) Mostly | (3) Mostly |
| | (4) Very Much | (4) Very Much |

Follow up questions for comprehension have the correct answer noted in bold. For enjoyment, understanding, and desire for updates, participants were only presented with the phrases "Not at all", "A bit", "Average", "Mostly", and "Very Much". The numbers in parentheses were added for analysis and presentation of data.

[a] The word 'reading' was removed or changed to 'viewing' for surveys with video or graphical abstracts.

[b] The word 'abstract' was changed to 'video summary' or 'summary' for surveys with video abstracts, 'summary' for surveys with plain language summaries, and 'graphical summary' or 'summary' for surveys with graphical abstracts.

## Survey analysis

Results were downloaded from Google Forms and put into Google Sheets for analysis. Results from surveys which showed the Takata et al. summary first and results from surveys which

showed the Cohn et al. summary first were compared via the Mann-Whitney U-Test to see if the populations were different based on which summary was presented first. In all cases of all summary types, it did not matter which summary was shown first, so Takata et al. data from both versions were pooled and Cohn et al. data from both versions were pooled (Fig 1).

The results were then checked between Cohn et al. and Takata et al. to see if the two papers yielded different results. The two papers showed statistically significant differences between the Cohn et al. and Takata et al. published abstracts in comprehension scores and the desire for more updates. Since the published abstracts were significantly different in these categories, the two papers were kept separate for analysis.

Participants who marked that they were undergraduates were pooled with participants who marked that they had non-science careers since undergraduates were in the minority of participants and they often have the same schooling as adults without science careers. There were no significant differences in the results between these two populations, so pooling seemed appropriate.

The results from the separate careers (science, science-related, and non-science) in the two papers were compared to see if there were significant differences. There were statistically significant differences between careers in every scoring category in each of the two papers, so the careers were kept separate for analysis.

All statistical significance between populations was calculated by the Mann-Whitney U Test [20]. All correlation values were calculated by the Pearson's r correlation coefficient.

## Results

### Survey participants and preferences

Participation in the survey was fairly even across careers. The Cohn et al. data set contained 201 science, 156 science-related, and 181 non-science participants (Table 3). The Takata et al. data set contained fewer science related (n = 112) and non-science (n = 133) participants due to a Google Form error that was corrected shortly after the survey was first publicized (Fig 1 and Table 3). Of the 538 total participants, 505 reported having a binary gender. The female/male split of those 505 participants was fairly even with the science and non-science participants having approximately a 60:40 split and the science-related participants having a 70:30 split. The 70:30 female/male split in science-related participants is representative of the number of women in outreach and other science-related careers as compared to men in those same careers [21].

The number of participants who viewed each type of summary is also fairly even. Only science-related video participants are lagging in number of participants (n = 26 for Cohn et al., n = 18 for Takata et al.) compared to the other summary and career types (n>39 for Cohn et al., n>29 for Takata et al.).

Before showing participants a science summary, our survey asked participants to report their preference for getting scientific information via written summaries, graphics/infographics, videos, audio sources like podcasts, and reading the original research paper. We gathered this information to see how much prior preferences affect the comprehension, enjoyment, understanding, and desire for updates of the different summary types. Participants from all careers had the same hierarchy of reported learning preferences with the exception of research papers (Fig 2 and S1 File). Written summaries were by far the most preferred learning type followed by graphics/infographics, videos, and then audio/podcasts (Fig 2). For research papers, non-science participants preferred them the least, science-related participants preferred them second only to written summaries, and science participants preferred them the most (S1 File).

**Table 3. Participant numbers for Cohn et al. and Takata et al.**

| Participant Numbers for Cohn et al. | | | | | |
|---|---|---|---|---|---|
| Career | Video | Graphic | Plain Language | Abstract | Total |
| Science | 42 | 49 | 49 | 61 | *201* |
| Non-Science | 44 | 47 | 47 | 43 | *181* |
| Science-related | 26 | 42 | 39 | 49 | *156* |
| Total | *112* | *138* | *135* | *153* | **538** |
| Participant Numbers for Takata et al. | | | | | |
| Career | Video | Graphic | Plain Language | Abstract | Total |
| Science | 42 | 49 | 49 | 61 | *201* |
| Non-Science | 35 | 31 | 38 | 29 | *133* |
| Science-Related | 18 | 29 | 30 | 35 | *112* |
| Total | *95* | *109* | *117* | *125* | **446** |

Numbers of participants separated by paper, by career, and by summary type.

We also wanted to know how scientists like to receive updates inside versus outside of their field so we could learn how that might affect their view of the different science summaries. We gathered information by asking participants with science careers how much they preferred getting research updates via scientific journals, newspaper articles, social media, recommendations from colleagues, or by PubMed alert. Recommendations from scientific journals, recommendations from friends and colleagues, and PubMed/Other Alerts are the most preferred update mechanisms inside the scientists' field of study (Fig 2B). When asked about their preferences outside their field, recommendations from friends and colleagues were the most preferred followed by social media, then newspaper articles and recommendations from scientific journals (Fig 2B).

## Video and plain language summaries are the most effective regardless of career

Given the clear reported preference for written forms of communication (Fig 2A), it was expected that the plain language summaries and perhaps the published abstracts would be the most effective summaries when tested. Contrary to our expectations, videos had the highest scores for comprehension, understanding, enjoyment, and desired updates (Median(M)>4 of 6, M>3 of 4, M>3 of 4, M>2 of 4 for videos respectively) (Fig 3). Plain language summaries often had equal scores to videos (M>4 of 6, M>2 of 4, M>2 of 4, M>2 of 4 for plain language respectively), but videos were either as effective or more effective than plain language summaries in all cases except comprehension of science-related participants for the Cohn et al. paper where plain language summaries had a higher average score (M = 4 of 6 for video, M = 5 of 6 for plain language) (Fig 3).

The differences in comprehension were generally small across summary types and careers. These small differences indicated that people are able to get the main takeaways of the paper no matter what type of summary they are shown. When statistically significant differences did occur, they indicated that video and plain language summaries produced higher comprehension scores (Fig 3).

Video and plain language summaries had higher reported understanding scores than published abstracts and graphical abstracts (Fig 3). This was true of participants from all careers and was true for both papers tested. In some cases, video even outperformed plain language

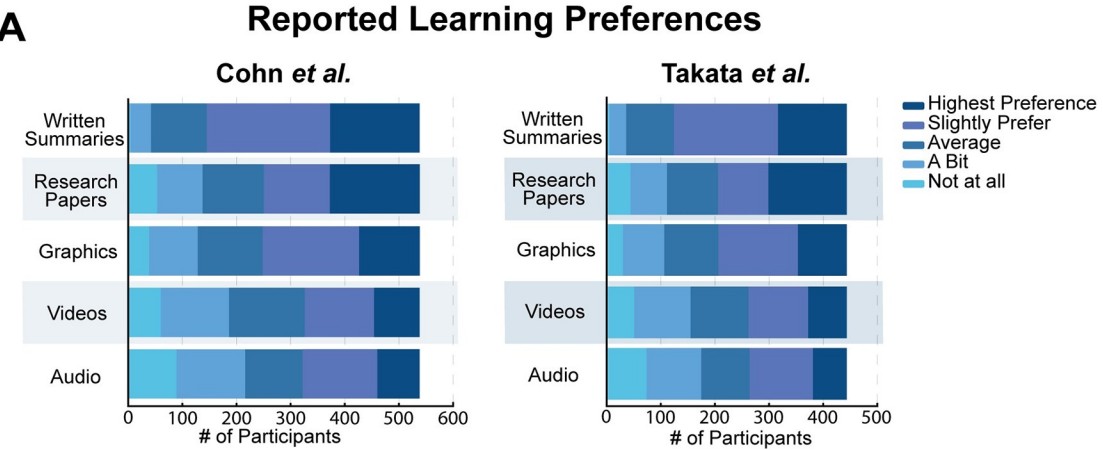

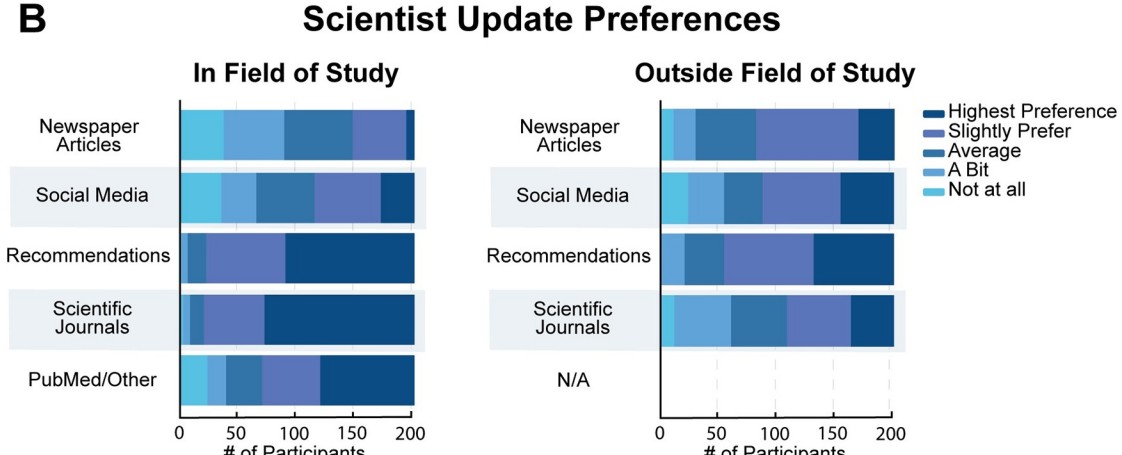

**Fig 2. Participant reported preferences.** Reported learning preferences for all participants and update preferences of participants with science careers. (**A**) shows data of all participants that answered the Cohn et al. paper and the Takata et al. paper. The bar charts show the reported preference of the participants for different ways to hear about science. (**B**) shows the update preferences of science participants both in their field of study and outside of it. The graph on the left shows preferences for research inside the scientist's field of study and the right shows preferences for research outside the field of study.

summaries (Fig 3), which is surprising given that the majority of participants ranked written summaries as their highest preference for getting new scientific information (Fig 2A).

Videos and plain language summaries had the highest enjoyment scores, but there were some differences between careers. Participants with science careers enjoyed videos the most (M = 4 of 4 for Cohn et al. and Takata et al., p<0.00001) followed by plain language summaries (M = 2 of 4 for Cohn et al., M = 3 of 4 for Takata et al., p<0.00001). Participants with non-science careers enjoyed videos the most as well (M = 3 of 4 for Cohn et al. and Takata et al., p<0.00001). Participants with science-related careers liked the videos and plain language summaries equally (M = 3 of 4 for video and plain language for Cohn et al. and Takata et al.). Published abstracts and graphical abstracts were enjoyed the least by all careers (all p<0.0027) with the exception of non-science participants who enjoyed abstracts the least (M = 1 of 4 for Cohn et al., M = 0 of 4 for Takata et al.), but enjoyed graphical abstracts and plain language summaries equally (M = 2 of 4 for graphic, M = 2 of 4 for plain language for Cohn et al.; M = 1 of 4 for graphic, M = 2 of 4 for plain language for Takata et al.) (Fig 3).

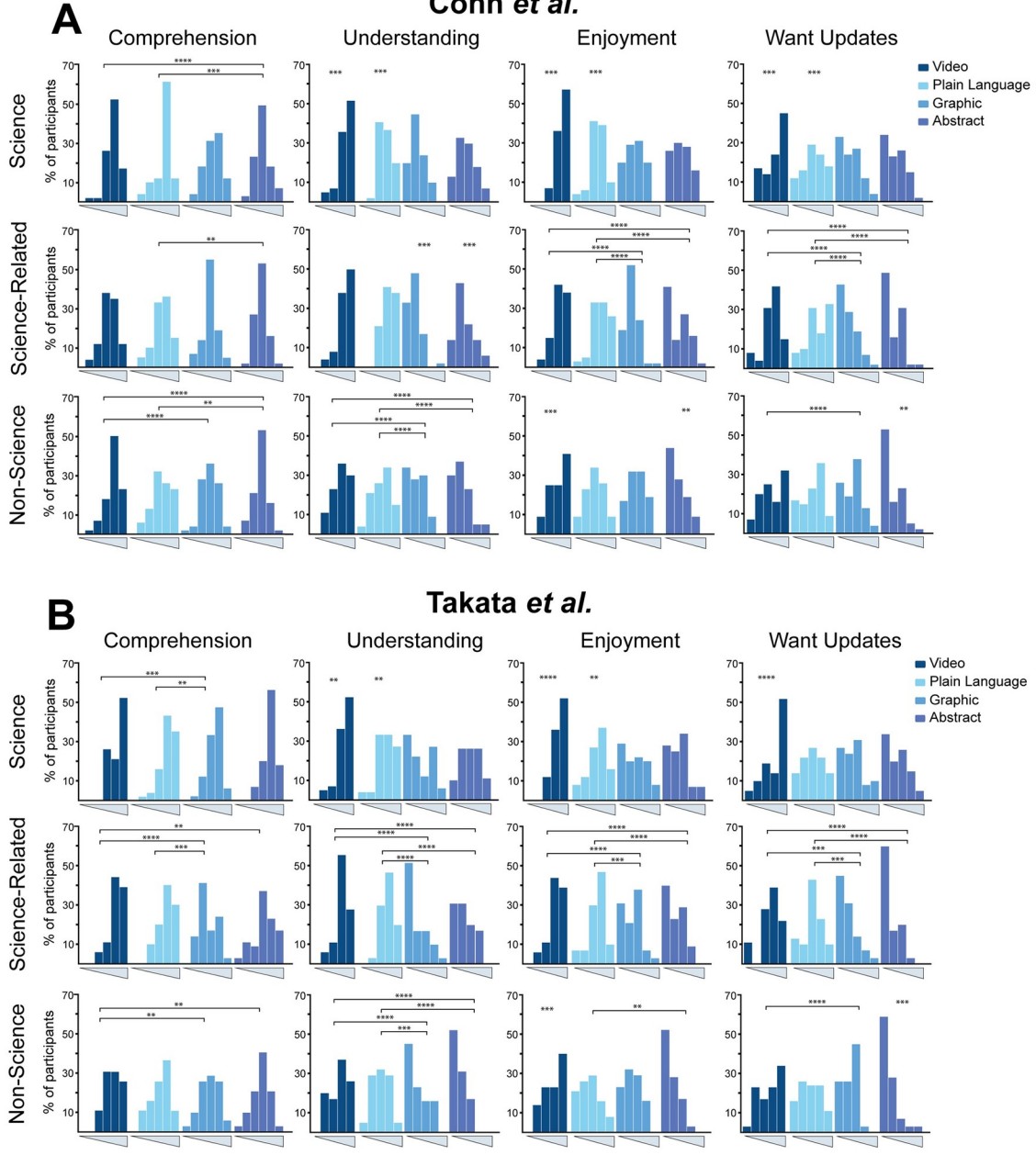

**Fig 3. All data from all summaries.** Histograms of the comprehension, understanding, enjoyment, and desire for more updates data for all survey types and all career types. **A** shows data for the Cohn et al. paper participants. **B** shows data for the Takata et al. participants. Each histogram shows the data as a percentage of participants. Comprehension histograms are plotted from 1–6, and understanding, enjoyment, and want updates plots are plotted from 0–4. Comprehension scores are from a series of questions asked in the survey (Table 2, S6 File). Understanding, enjoyment, and want updates scores are numerical representations of responses where 0 was "not at all" and 4 was "very much" (Table 2, S6 File). Statistical significance is shown above each plot where p<0.01 using the Mann-Whitney U-Test. Specifically, the asterisks represent the following p-values: p<0.00001(****), p<0.0001(***), p<0.001(**), p<0.01(*).

When asked if they wanted to get more updates in the form of the summary they saw, participants rated videos or plain language summaries the highest, independent of career (all p<0.00148 for video, all p<0.01 for plain language summaries) (Fig 3). Published abstracts and graphical abstracts had the lowest update scores (all p<0.00148) with the exception of

non-science participants who scored published abstracts the lowest (M = 0 for Cohn et al. and Takata et al.), but scored graphics and plain language summaries equally (M = 2 of 4 for Cohn et al.; M = 1 of 4 for graphic, M = 2 of 4 for plain language for Takata et al.) (Fig 3).

Overall, video abstracts and plain language summaries produced the highest comprehension, understanding, enjoyment, and desire for more updates. This led us to the conclusion that video abstracts and plain language summaries are the most effective summary formats regardless of career.

## Strong correlations exist between reported learning preferences and summary ranks

Generally participants from all careers felt similarly about the summaries. All participants scored the video and plain language summaries the highest and the graphical abstracts and published abstracts the lowest in all categories. We thought that if we sorted the participants by their reported preferences rather than by their careers, we might see strong correlations between reported preference and comprehension, understanding, enjoyment, or desire for updates.

To see whether reported preference was correlated with the summary scores, we looked at the comprehension, enjoyment, understanding, and update scores from participants that viewed each of the summaries and saw if those scores correlated with their reported preference for getting updates of that type. For example, to look at the published abstracts, we looked to see if the comprehension, understanding, enjoyment, and desire for updates scores correlated with the reported preference for reading the original research paper. For the video scores, we looked to see if the scores correlated with the video reported preference. For the graphical abstract scores, we looked to see if the scores correlated with the graphic/infographic preference. We could not evaluate plain language summaries because almost all of the participants marked average or higher preference for written summaries when asked how they prefer to get science updates prior to viewing any of our summaries (Fig 2A). This limited our ability to see any correlation, so plain language summaries were not analyzed. Videos, graphical abstracts, and published abstracts each had a wider distribution of reported preferences from lowest to highest, so they were analyzed (Fig 2A).

Although the comprehension, understanding, enjoyment, and update scores were similar between the Takata et al. and Cohn et al. papers when the data was separated by career, the preference correlations showed a clear difference between the two papers (Fig 4). In the Cohn et al. data set, comprehension score was not correlated with the reported preference in any of the summary types (Fig 4A). This lack of correlation means that participants did not perform better on the comprehension test when they were paired with their preferred type of summary whether it was a video, graphic, or the original published abstract. The Takata et al. data showed similar results for the video and graphical summaries, but it showed a significant correlation between preference for reading the original research paper and the published abstract comprehension score (r = 0.29, p = 0.0009) (Fig 4B). This correlation indicates that participants which marked reading the original research paper as their highest preference performed better on the comprehension test and participants that marked reading the original research paper lower performed worse. This Takata et al. specific correlation could be due to the basic biology nature of that paper and the background knowledge required to understand their findings.

Significant correlations were also seen for published abstracts in reported understanding, enjoyment, and the desire for more updates in both papers (all p<0.00001). This indicates that participants which prefer reading the original research paper also score the abstracts higher in

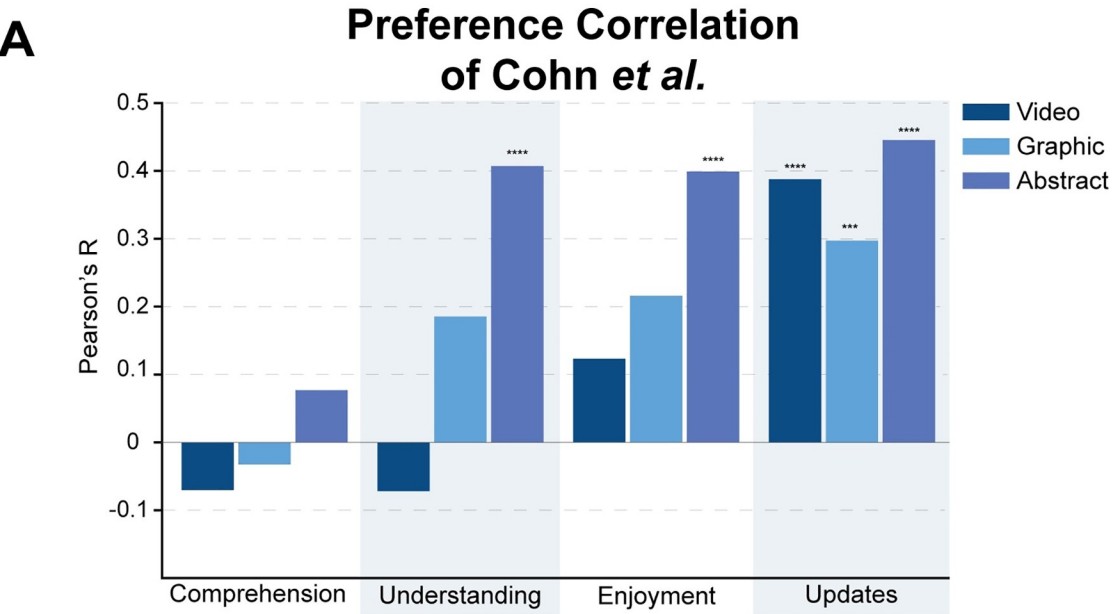

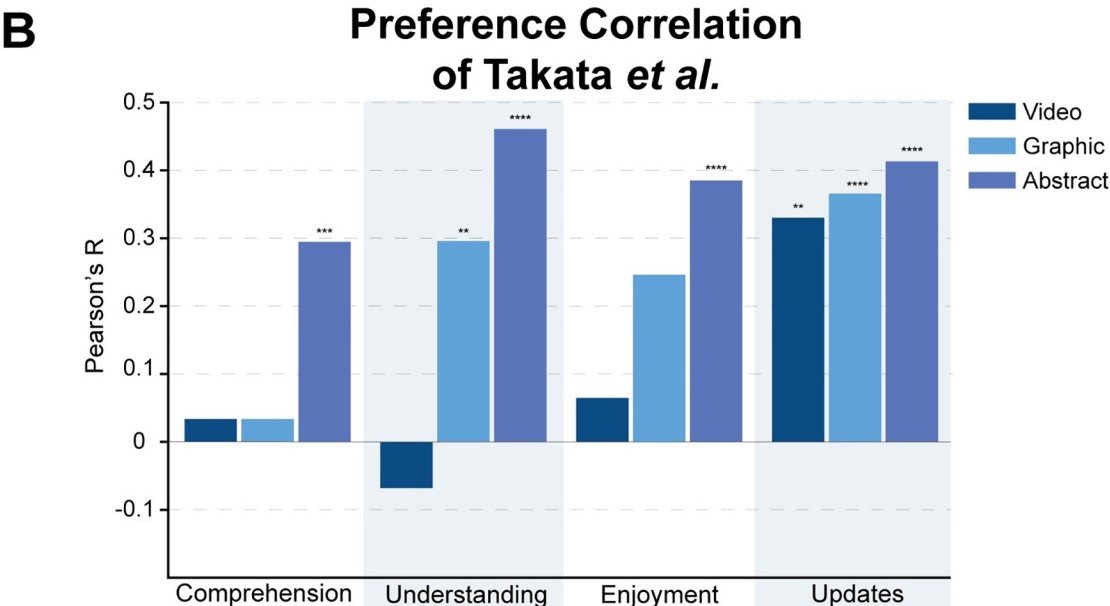

**Fig 4. Correlations between reported preference and summary values.** Bar graphs of preference correlation for Cohn et al. and Takata et al. papers. Both graphs show data for videos, graphics, and published abstracts. Analysis was not completed for plain language summaries due to the overwhelming reported preference for written summaries (see Fig 2 for reported preference data). For each summary type, the reported preference for that type was tested for correlation with the comprehension score, reported understanding, reported enjoyment, or the desire for more updates of that type using a Pearson's r correlation calculation. **A** shows the data for Cohn et al. **B** shows the data for Takata et al. Statistical significance is noted where p<0.01. Specifically, the asterisks represent the following p-values: p<0.00001(****), p<0.0001(***), p<0.001(**), p<0.01(*).

all categories, and those that do not prefer reading the original research paper score the abstracts lower in all categories. The same correlations were not seen for videos. The only significant correlation for the video summaries was a correlation between the desire for more video updates and the reported video preference (r = 0.387, p = 0.000024 for Cohn et al.;

r = 0.33, p = 0.0011 for Takata et al.) (Fig 4). This indicates that participants who reported a preference for videos wanted to keep seeing more videos even after viewing our video abstracts. The lack of correlation between video reported preference and understanding/enjoyment highlights how effective videos were overall. Participants gave high scores to the video abstracts in the understanding and enjoyment categories regardless of whether they reported that they preferred videos as a way to get new scientific information before seeing our video abstracts.

### Reported understanding and comprehension show strong correlations for Takata et al. summaries

It might be expected that the better you perform on a quiz, the more confident you are that you understood the material covered in that quiz. When we examined the relationship between comprehension score and reported understanding, the Cohn et al. and Takata et al. papers diverged (Fig 5).

The Cohn et al. data showed correlations between comprehension scores and understanding scores for the video (r = 0.223, p = 0.018) and plain language summaries (r = 0.193, p = 0.025), but no correlation for the graphical abstracts or published abstracts (Fig 5). The correlation between understanding and comprehension scores in video and plain language summaries suggest that participants felt confident in their answers and their understanding of the Cohn et al. paper after watching the video or reading the plain language summary. It also suggests that participants did not feel as confident after reading the published abstract or viewing the graphical abstract.

Contrary to the Cohn et al. data, the Takata et al. data showed significant correlations in all summary types (all p<0.00018) (Fig 5). These correlations hint at the possibility that more background knowledge is needed to understand the findings of the Takata et al. paper as compared to the Cohn et al. paper.

## Discussion

We found that videos and plain language summaries are the most effective summaries, based on comprehension, understanding, and enjoyment. This finding was independent of an individual's career type. Surprising to us, our results also hint that there can be differences between papers based on the background knowledge required to understand the findings of that paper even when the readability of the two paper summaries are similar.

The Takata et al. paper shows that HIV-1 has selectively removed CG dinucleotides from its genome to more closely mimic the nucleotide content of its human host [15]. Understanding this finding requires a solid understanding of DNA and basic molecular biology concepts. In contrast, the Cohn et al. paper outlines a method for recovering latent cells from HIV-1 + patient blood in order to study them for a possible future cure [14]. This paper doesn't require nearly the depth of molecular biology understanding as the Takata et al. paper, and that difference seems to show up in the data.

Individuals with a strong science background are more likely to report that they prefer to get information from scientific papers since they are more likely to still be working in academia and reading scientific papers on a regular basis. That preference for scientific papers then correlates strongly with the comprehension score, but only for the Takata et al. paper where background knowledge is necessary to understand the finding (r = 0.3, p = 0.0009) (Fig 4B). It would be illuminating to create a survey with a broader range of papers to see if this trend holds true.

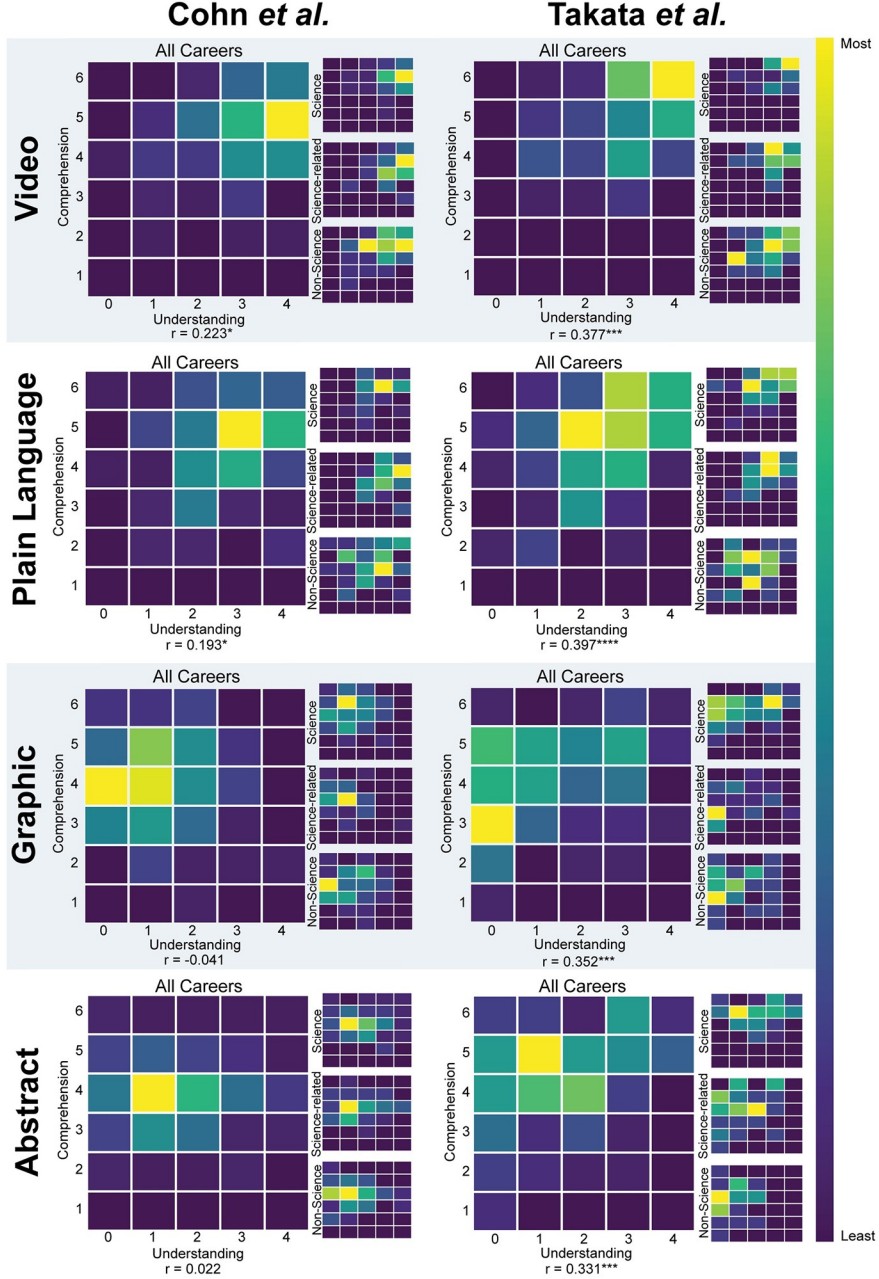

**Fig 5. Heat maps of reported understanding versus comprehension score.** Heat maps of reported understanding versus comprehension score of Cohn et al. and Takata et al. separated by summary type. The larger heat maps show the summed data for all participants and the three smaller heat maps to the right show the data for each career type. Each larger heat map contains the Pearson's r correlation value for all careers. Statistical significance is noted where p<0.05. Specifically, the asterisks represent the following p-values: p<0.00005(****), p<0.0005(***), p<0.005(**), p<0.05(*).

In addition to looking at a broader range of papers, we would also like to see more research into how the quality of the summary is related to the effectiveness of that summary. The quality of the content of published plain language summaries, video abstracts, and graphical abstracts is known to vary despite the rigorous technical specifications and instructions outlined by the journals that require these summaries [6,16–18]. Video abstracts and graphical abstracts are

produced by the researchers themselves, and not every research team has the skills necessary to create these summaries. Occasionally, videos will be made by a company that the researchers hire, like Animate Your Science. However, this results in a greater variability in quality of the video abstracts since professional videographers generally have access to better film equipment and editing software than the average researcher making a video. Plain language summaries also differ from journal to journal as some are written by editors and other by the authors themselves [3].

Previous research on plain language summaries suggests that multiple rounds of editing and collaborations with members of your intended audience can help make better summaries [22,23]. Perhaps this would also be true for videos and graphical abstracts, but more research on this area is needed.

Overall, our research only identifies which science summary is the most effective when a single researcher carefully creates content-identical summaries following the rules set out by the journals that are associated with those summaries. There is still much more work to be done in order to know exactly how we should summarize our science. One type of summary that was completely omitted in this study was podcasts. Cell, Science, and Nature all produce podcasts about some of the most relevant research that they publish. It is unclear how a podcast would perform relative to videos, graphics, and plain language summaries.

This research also only included graphical abstracts that closely mimic the currently published Cell graphical abstracts, following the best practices that Cell has laid out [6]. It's possible that infographics could be more successful than the current graphical abstracts. Infographics function as a combination of word descriptions and images together. They often include data and small descriptions of the work, and they are meant to inform the viewer on a particular topic. The current graphical abstracts encourage the creators of the graphics to avoid using any sentences and to avoid adding any data to the image [6]. Participants of this survey often commented on the graphical abstracts saying that they wished they had words or even a small description to go with the image which suggests that infographics might be more helpful at conveying information.

Cell has stated that their graphical abstracts are meant as almost an advertising tool to encourage readers to browse papers and see which ones might be interesting to them whether they are in that scientist's field of interest or outside of it [6]. Perhaps the graphical abstracts performed poorly in this research simply because they are not meant to provide the key takeaways of the research article in the same way that videos and plain language summaries are. Given the time involved in making these graphical abstracts, our results suggest that the time might be better invested in preparing an alternative, such as a video or a plain language summary, if the goal is to help interest readers in other articles.

We also recommend more research on video abstracts specifically, since they were the most effective in our analysis. There is a slight negative, though not statistically significant, correlation of video preference and reported understanding in both papers (r = -0.07 for Cohn et al. and Takata et al.), which indicates that although many people did not report a preference for videos, they understood the research much better after viewing them (Fig 4). The enjoyment of the videos was also not strongly correlated with video preference even though people reported high enjoyment scores after watching the videos (Figs 3 and 4). A comments section was available at the end of the surveys and the comments for the videos were very positive and many participants indicated that they were surprised at how helpful a video abstract could be.

Despite videos being the most effective, we recognize that there are limitations to their implementation due to the inherent cost and time involved. Not every researcher has the resources available to produce a video summary. Also, participants did not report that they preferred videos more than written summaries or graphics before viewing our summaries

(Fig 2A), so recruiting a large number of people to watch a video might be challenging even though it has been shown here to be the most effective.

Based on this study, we suggest that all researchers consider writing a plain language summary of their research. Those summaries can be published with their paper or published in other relevant locations including lab websites, university websites, or university newsletters. To get started with a plain language summary, it can be helpful to look at the eLife questions for researchers or the Cochrane Methods for writing a plain language summary [16,24]. Also, we recommend editing the summaries at least once, hopefully after getting feedback from a member of the intended audience and possibly using a jargon detection program to make sure the summary is accessible [22,25]. Summaries can be put into a readability calculator to help make sure that the summary is easy to read. However, we don't recommend depending solely on readability scores because scientific summaries must use unfamiliar words or phrases at times for accuracy, and readability calculators only report on how easy a written document is to read, not how easy it is to understand. Finally, a plain language summary can be a great way to organize a research paper as it forces researchers to focus on the take-home message, so writing plain language summaries can have benefits to both the researcher and to the people the research is trying to communicate with.

If the findings of their research are of public relevance, researchers could consider investing the time and money into a video of their results. Videos had the highest ratings across the board and they left people feeling very confident and positive about the research being presented (Fig 3). Not all papers require a video, but it is an excellent option for select relevant papers that should not be overlooked.

We also recommend that journals consider including plain language summaries with all of their papers as a separate section available outside the paywall, if a paywall exists for that publication. We also recommend that journals with topics of high public relevance or those that would benefit from strong interdisciplinary ties consider creating videos of their papers to share with the community.

## Supporting information

**S1 File. Reported learning preferences by career for Cohn et al.** The bar charts show the reported preference of the participants for different ways to hear about science separated by career for the Cohn et al. data set.
(TIF)

**S2 File. Cohn et al. video abstract.** A compressed version of the video created for Cohn et al. paper. The video script used was the same as the plain language summary. The video was created with Autodesk Sketchbook, GarageBand, and iMovie software and was hosted on YouTube. The video was embedded into the survey for participants to view. The closed captioning was edited to be on with the option for the user to turn it off. For the full video see: youtu.be/RLuunA81kJo.
(MP4)

**S3 File. Takata et al. video abstract.** A compressed version of the video created for Takata et al. paper. The video script used was the same as the plain language summary. The video was created with Autodesk Sketchbook, GarageBand, and iMovie software, and was hosted on YouTube. The video was embedded into the survey for participants to view. The closed captioning was edited to be on with the option for the user to turn it off. For the full video see: youtu.be/Kp-0PvS99fM.
(MP4)

S4 File. Graphical abstracts. Graphical abstracts created for the Cohn et al. (**A**) and Takata et al. (**B**) papers. Graphical abstracts used similar visual motifs as the video abstracts and were created using Keynote software. Each abstract was put through a color blindness simulator to ensure that the abstracts could be seen properly by all viewers. The abstracts were embedded into the survey for participants to review.
(TIF)

S5 File. Plain language summaries. Plain language summaries written for the Cohn et al. (**A**) and Takata et al. (**B**) papers. Summaries were written based on intensive review of the published papers. The summaries also hit each key point mentioned in the abstracts of each paper. The Cohn et al. summary contains 422 words (**A**) and the Takata et al. summary contains 433 words (**B**). Each summary was embedded into the survey for participants to review.
(TIF)

S6 File. Copy of published abstract survey. A PDF copy of the survey presented to participants. This copy shows the published abstracts as the summary type. It has the Cohn et al. summary shown first and the Takata et al. shown second. Other surveys are identical to this one except that they show videos, plain language summaries, or graphical abstracts instead of the published abstracts. The videos can be seen in S2 and S3 Files. The graphical abstracts are in S4 File and the plain language summaries are in S5 File. Half of the surveys have the Cohn et al. summary shown first and half have the Takata et al. summary shown first. (See Fig 1 for schematic).
(PDF)

S7 File. Survey data. All data from the survey.
(XLSX)

# Acknowledgments

We thank Dr. Michelle Itano (University of North Carolina) and Dr. Jeanne Garbarino (Rockefeller University) for useful discussions. We also thank Arlene Hurley (Rockefeller University) for help with our IRB application preparation.

# Author Contributions

**Conceptualization:** Kate Bredbenner, Sanford M. Simon.

**Data curation:** Kate Bredbenner.

**Formal analysis:** Kate Bredbenner.

**Investigation:** Kate Bredbenner.

**Methodology:** Kate Bredbenner.

**Project administration:** Kate Bredbenner.

**Supervision:** Sanford M. Simon.

**Validation:** Kate Bredbenner.

**Visualization:** Kate Bredbenner.

**Writing – original draft:** Kate Bredbenner.

**Writing – review & editing:** Kate Bredbenner, Sanford M. Simon.

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
