## [Decision Letter · Decision Letter 0]

10 Jul 2019

PONE-D-19-16933

Scientists should use plain language summaries and video abstracts to summarize their research

PLOS ONE

Dear Ms Bredbenner,

Thank you for submitting your manuscript to PLOS ONE. After careful consideration, we feel that it has merit but does not fully meet PLOS ONE’s publication criteria as it currently stands. Therefore, we invite you to submit a revised version of the manuscript that addresses the points raised during the review process.

Many thanks for your submission. This is a very interesting and pertinent topic which can certainly be published in the special issue. However, considering the comments from the reviewers, I would highly recommend you re-structure the article to present the methods before the results and clarify the procedure you used. Please follow the recommendations of the reviewers in your revision.

We would appreciate receiving your revised manuscript by Aug 24 2019 11:59PM. To enhance the reproducibility of your results, we recommend that if applicable you deposit your laboratory protocols in protocols.io, where a protocol can be assigned its own identifier (DOI) such that it can be cited independently in the future. For instructions see: http://journals.plos.org/plosone/s/submission-guidelines#loc-laboratory-protocols

We look forward to receiving your revised manuscript.

Kind regards,

David Orrego-Carmona, Ph.D.

Academic Editor

PLOS ONE

Journal Requirements:

2. Could you please include a copy of the questionnaire as Supplementary Information? Currently the questions asked are only available in the data files and hard to extract for the reader.

I have read the journal's policy and the authors of this manuscript have the following competing interests: KB is a paid creator of video abstracts via SimpleBiologist. https://www.simplebiologist.com

Reviewers' comments:

Reviewer's Responses to Questions

**Comments to the Author**

1. Is the manuscript technically sound, and do the data support the conclusions?

Reviewer #1: Yes

Reviewer #2: Yes

2. Has the statistical analysis been performed appropriately and rigorously? 

Reviewer #1: Yes

Reviewer #2: Yes

3. Have the authors made all data underlying the findings in their manuscript fully available?

Reviewer #1: Yes

Reviewer #2: Yes

4. Is the manuscript presented in an intelligible fashion and written in standard English?

Reviewer #1: Yes

Reviewer #2: Yes

5. Review Comments to the Author

Reviewer #1: This is an interesting and important topic. The methods were appropriate, but I think the synthesis and presentation of results could be improved to make the paper more accessible to readers.

I found the abstract and introduction clearly written, although the introduction was a little long. Other sections were very difficult to follow. Having the methods at the end of the paper was extremely confusing and made the paper very difficult to read – you need to understand the methods used to be able to interpret the results. The results seemed overly complicated and very difficult to follow.

Given that your focus is on the benefits of alternative models of presentation, how about providing a video summary and PLS of this paper?

Major suggestions

Some of the methods are rather confusing. I can’t work out whether each participant viewed a summary for each of the two papers and if so, whether they viewed the same type of summary. Could you reword to make this clear? It does become clear at line 500 but should be made explicit from the start. Actually, having moved on to line 504 I am now confused again! This suggests that they did receive summaries for two papers. I think I have worked out now that you have 8 surveys because the order in which the papers are presented is switched but this isn’t immediately clear – please simplify this section so it is clear what information each participant received.

I’m not sure of the purpose of the additional bit of the survey for those with science careers. Given that this paper is already very difficult to follow, and that this is not the main objective, perhaps it might be clearer to remove this?

I would like more information on the questions asked to determine comprehension. Could these be described in more detail somewhere? Perhaps in a table?

I think you are possibly over-analysing all the different subgroups given the relatively small numbers. It may be better to focus on overall results first and then discuss any differences between your groups of participants.

I question the value of the correlation analysis. This is confusing and difficult to follow and I’m not sure it adds much to the take home message of the paper.

I would like to see some more detail on how outcomes were measured. You have rating preferences on the figures, but I don’t quite understand these. I presume participants were asked to rate on 5-point scale but what does “slightly prefer” and “a bit” mean? Prefer to what?

Some data would be easier to follow in tables – e.g. summary of participants. Some of the detail could then be removed from the text and figures to make it easier to follow.

The way results are described is really confusing. You talk about participants within each data set. But the datasets don’t have participants (the survey does) and they are summaries of research papers not data sets. Could you consider clearer wording?

“Surprising to us, the results also show that there can be differences between papers based on the background knowledge required to understand the findings of that paper.” – I’m not quite sure of the basis for this claim and wonder whether you are reading too much into an analysis based only on two different papers?

The discussion describes what the two papers you were looking at. I think a box or table in the methods would be more appropriate for this – this would be helpful information when people are considering what you have done.

I wonder in the discussion whether you might consider adding something on the benefits/costs to researchers in writing a PLS or producing a video summary. For example, I find that to write a PLS for my research I need to really think carefully about what the take home message is and what I want to say. This often leads to me improving other sections of the report based on the thinking/work I do to write a PLS. On the other hand, you may consider the extra resources needed to produce a video summary. I really liked your video summaries but I wouldn’t know how to go about producing one.

Minor suggestions

In the introduction you could also add the all Cochrane systematic reviews are required to produce a plain language summary.

Line 186 – “Video and PLS…” I don’t quite understand why this is here and written in bold

Line 528… Significant differences in what?

Line 551 “Normalisation…” Would it not be simpler to state that you used % rather than actual numbers for the histograms?

Line 166 (and elsewhere) “Trends in….” – you have done a cross-sectional survey so how can you comment on trends? I don’t understand this sentence.

Figure 1 – doesn’t really seem to be about demographics. It’s summarising your three groups of participants and their learning preferences. This is quite difficult to read.

Reviewer #2: The manuscript is really well written, and it is of great quality. The subject is important for the scientific field and it should be spread up so our research could reach all kinds of public.

Cochrane has a standard summary plain language format perhaps it is a good idea to encourage the writers to follow one model until papers do not reach a consensus.

About the methods, I don’t know if it is preference of the authors but the methods section after the results makes the understanding a little bit difficult.

Also, I am not sure if it is clear about how the preference for the type of abstract was assessed. By the text “Participants were randomly assigned to one of the eight possible surveys via a random URL generator embedded into the button on the survey website. No single participant ever saw more than one of the different summary types.”, so each participant only saw one type of summary (video, graphical, plain language and published abstracts). If each participant only saw one type of abstract (both Takata and Cohn) how do you know which the person prefers?

Also, you report the number of people that saw each article “The Takata et al. data set contains fewer science related (n=112) and non-science (n=133)”, but how many saw each type of abstract of each author?

I believe this matter should be enlightened for the final version.

6. PLOS authors have the option to publish the peer review history of their article (what does this mean?). If published, this will include your full peer review and any attached files.

Reviewer #1: No

Reviewer #2: No

---

## [Author Response · Author response to Decision Letter 0]

5 Aug 2019

Reviewer’s Comments

Reviewer #1:

1. Some of the methods are rather confusing. I can’t work out whether each participant viewed a summary for each of the two papers and if so, whether they viewed the same type of summary. Could you reword to make this clear? It does become clear at line 500 but should be made explicit from the start. Actually, having moved on to line 504 I am now confused again! This suggests that they did receive summaries for two papers. I think I have worked out now that you have 8 surveys because the order in which the papers are presented is switched but this isn’t immediately clear – please simplify this section so it is clear what information each participant received.

The methods section has been rewritten to be as clear as possible and a new figure was added that shows a flowchart of how participants were funneled into the 8 different surveys (fig 1).

2. I’m not sure of the purpose of the additional bit of the survey for those with science careers. Given that this paper is already very difficult to follow, and that this is not the main objective, perhaps it might be clearer to remove this?

We designed this section of the survey because, although it would be incredible if people from all walks of life read scientific journals, the majority of journal traffic comes from people with science careers. We thought it was a good idea to get information about where scientists are already going to hear about updates inside and outside of their fields so we can try to create summaries that work with where scientists already are and what they already use. We have updated the results section where we talk about this information to better reflect our intentions with these questions.

3. I would like more information on the questions asked to determine comprehension. Could these be described in more detail somewhere? Perhaps in a table?

The comprehension questions are now a part of a supplementary figure that displays the survey (S6 File). There is also a table of the comprehension questions in the methods section of the manuscript (Table 1).

4. I think you are possibly over-analysing all the different subgroups given the relatively small numbers. It may be better to focus on overall results first and then discuss any differences between your groups of participants.

Since we found that there were statistically significant differences between our sub-groups, we didn’t feel comfortable pooling them. We have gone through the results section and stated the overall outcome before discussing differences between subgroups to improve clarity. 

5. I question the value of the correlation analysis. This is confusing and difficult to follow and I’m not sure it adds much to the take home message of the paper.

We believe that this analysis does show important aspects of our data including the differences that show up because of how you prefer to learn or get updates. This is an important distinction and we feel that the correlation helps us present that well. Changes have been made in the results section that should emphasize the importance. 

6. I would like to see some more detail on how outcomes were measured. You have rating preferences on the figures, but I don’t quite understand these. I presume participants were asked to rate on 5-point scale but what does “slightly prefer” and “a bit” mean? Prefer to what?

For figure 2, we asked the participants to rank the way that they like to hear about new science (before ever seeing the summaries we were testing), so the preferences are between the different options for getting information (video, audio, written, etc). For the other figures, there is not a preference but instead participants say how much they agree with statements like “I enjoyed reading this abstract” or “I understand this research more after reading this abstract.” Hopefully the addition of the survey as a supplementary file (S6 File) and the additional changes to the methods help clarify this issue (Table 1). 

7. Some data would be easier to follow in tables – e.g. summary of participants. Some of the detail could then be removed from the text and figures to make it easier to follow.

The Demographics Figure (prev. fig 1) has been split into a table (Table 2) that shows the participant numbers and a figure (fig 2) which shows the reported preferences of the participants. 

8. The way results are described is really confusing. You talk about participants within each data set. But the datasets don’t have participants (the survey does) and they are summaries of research papers not data sets. Could you consider clearer wording?

We have edited the results section to be clearer on this point. 

9. “Surprising to us, the results also show that there can be differences between papers based on the background knowledge required to understand the findings of that paper.” – I’m not quite sure of the basis for this claim and wonder whether you are reading too much into an analysis based only on two different papers?

When we designed the survey, we chose a paper with a medical focus and one with a basic biology focus on purpose to be able to see if they showed the same results. Since we saw a difference between the two, especially in participants with science-related careers and non-science careers, we thought that this was something worth noting. We hope that someone will perform additional research which addresses this question more fully than we have in this paper, but we think it is worth discussing. We changed the wording of this paragraph to better reflect that our results are hinting at this possibility rather than showing it explicitly. 

10. The discussion describes what the two papers you were looking at. I think a box or table in the methods would be more appropriate for this – this would be helpful information when people are considering what you have done.

The methods have been updated to more fully cover the scope of the survey and the two papers. 

11. I wonder in the discussion whether you might consider adding something on the benefits/costs to researchers in writing a PLS or producing a video summary. For example, I find that to write a PLS for my research I need to really think carefully about what the take home message is and what I want to say. This often leads to me improving other sections of the report based on the thinking/work I do to write a PLS. On the other hand, you may consider the extra resources needed to produce a video summary. I really liked your video summaries but I wouldn’t know how to go about producing one.

We did have some consideration of the benefits/costs for videos and PLS in the discussion, but we have added more based on this comment. Previously we focused on time and cost. Now we have also included the ideas suggested in this comment. 

12. In the introduction you could also add the all Cochrane systematic reviews are required to produce a plain language summary.

This information has been added to the introduction (line 72).

13. Line 186 – “Video and PLS…” I don’t quite understand why this is here and written in bold

“Video and Plain Language Summaries are the most effective regardless of Career” is the title of that section of the results. We split the results into several sections for clarity. 

14. Line 528… Significant differences in what?

The methods have been rewritten to be clearer. In this case, we checked for significant differences between comprehension, enjoyment, understanding, and the desire for more updates between the two papers in each summary type and with each career. For example, we looked at whether there was a significant difference between the Cohn et al. and Takata et al. results from non-science participants that saw the video summary. The only significant differences we saw between papers were with the published abstract. 

15. Line 551 “Normalisation…” Would it not be simpler to state that you used % rather than actual numbers for the histograms?

This is a fair point. We thought normalization to 100 for each seemed simpler since every subgroup had different numbers of people, but we can say percentage instead. It has been changed in the final manuscript. 

16. Line 166 (and elsewhere) “Trends in….” – you have done a cross-sectional survey so how can you comment on trends? I don’t understand this sentence.

We were trying to use the “trends” phrase to make conclusions across subgroups. We have adjusted our language into something that we believe is clearer. 

17. Figure 1 – doesn’t really seem to be about demographics. It’s summarising your three groups of participants and their learning preferences. This is quite difficult to read.

We have adjusted figure one and it is now a table (table 2) and a figure (fig 2). We hope that makes the results clearer. 

Reviewer #2

1. Cochrane has a standard summary plain language format perhaps it is a good idea to encourage the writers to follow one model until papers do not reach a consensus.

We have added the fact that Cochrane systematic reviews have plain language summaries in our introduction (line 72), but after searching their website for some time, we could not find a good outline of how to write or format a plain language summary, so we did not include the suggestion that authors should follow the Cochrane format. 

2. About the methods, I don’t know if it is preference of the authors but the methods section after the results makes the understanding a little bit difficult.

We have updated the methods both in language and in location. It’s now before the results section.

3. Also, I am not sure if it is clear about how the preference for the type of abstract was assessed. By the text “Participants were randomly assigned to one of the eight possible surveys via a random URL generator embedded into the button on the survey website. No single participant ever saw more than one of the different summary types.”, so each participant only saw one type of summary (video, graphical, plain language and published abstracts). If each participant only saw one type of abstract (both Takata and Cohn) how do you know which the person prefers?

The reported preferences that are in figure 1 (now fig 2) were obtained by asking the participants how they prefer to get new scientific information (before ever seeing one of our created summaries). All other figures are based on asking participants to say how much they “enjoyed reading this abstract” or “understand this research more after reading this abstract.” Then the scores for each summary type were compared. We hope that the updates in the methods section (Table 1) and the supplemental figure (S6 File) that shows the survey will clear up this confusion. 

4. Also, you report the number of people that saw each article “The Takata et al. data set contains fewer science related (n=112) and non-science (n=133)”, but how many saw each type of abstract of each author?

I believe this matter should be enlightened for the final version.

We have separated the information from figure 1 into a table of the participant numbers (Table 2) and the reported preferences as a figure (fig 2). All Takata et al. participants saw the summaries by both authors. A certain portion of Cohn et al. participants only saw the Cohn et al. summary (the difference between the numbers of participants between Cohn and Takata). The table presentation of the participant numbers along with the updates to the methods should clear up this confusion.

---

## [Decision Letter · Decision Letter 1]

23 Sep 2019

PONE-D-19-16933R1

Video abstracts and plain language summaries are more effective than graphical abstracts and published abstracts

PLOS ONE

Dear Ms Bredbenner,

Thank you for submitting your manuscript to PLOS ONE. After careful consideration, we feel that it has merit but does not fully meet PLOS ONE’s publication criteria as it currently stands. Therefore, we invite you to submit a revised version of the manuscript that addresses the points raised during the review process.

We would appreciate receiving your revised manuscript by Nov 07 2019 11:59PM. To enhance the reproducibility of your results, we recommend that if applicable you deposit your laboratory protocols in protocols.io, where a protocol can be assigned its own identifier (DOI) such that it can be cited independently in the future. For instructions see: http://journals.plos.org/plosone/s/submission-guidelines#loc-laboratory-protocols

We look forward to receiving your revised manuscript.

Kind regards,

David Orrego-Carmona, Ph.D.

Academic Editor

PLOS ONE

Additional Editor Comments (if provided):

Dear authors,

Many thanks for the revised version of the manuscript. Please find attached some minor comments by the reviewer. I am enquiring about the possibility to add a video.

All the best,

David

Reviewers' comments:

Reviewer's Responses to Questions

**Comments to the Author**

1. If the authors have adequately addressed your comments raised in a previous round of review and you feel that this manuscript is now acceptable for publication, you may indicate that here to bypass the “Comments to the Author” section, enter your conflict of interest statement in the “Confidential to Editor” section, and submit your "Accept" recommendation.

Reviewer #2: All comments have been addressed

2. Is the manuscript technically sound, and do the data support the conclusions?

Reviewer #2: Yes

3. Has the statistical analysis been performed appropriately and rigorously? 

Reviewer #2: Yes

4. Have the authors made all data underlying the findings in their manuscript fully available?

Reviewer #2: Yes

5. Is the manuscript presented in an intelligible fashion and written in standard English?

Reviewer #2: Yes

6. Review Comments to the Author

Reviewer #2: This manuscript has a wide range of information and is a little bit difficult to follow. The changes made in the first review solved many of these problems and made it really easier and organized. That said, I noted a few more things that could be adjusted in this version once I could get a better idea of the message you are trying to send.

7. PLOS authors have the option to publish the peer review history of their article (what does this mean?). If published, this will include your full peer review and any attached files.

Reviewer #2: No

---

## [Author Response · Author response to Decision Letter 1]

8 Oct 2019

Reviewer Comments

1. If you go to Cochrane Methods you can find a section on Methodological Expectations of Cochrane Intervention Reviews (MECIR) (https://methods.cochrane.org/methodological-expectations-cochrane-intervention-reviews). There you will find a link to the Standards for the reporting of plain language summaries of new reviews of interventions by the Plain Language Expectations for Authors of Cochrane Summaries (PLEACS Initiative). (https://methods.cochrane.org/sites/default/files/public/uploads/pleacs_2019.pdf)

We have reviewed the Cochrane standards and found aspects of the methods that seem like good practices, so we have added the reference to the manuscript (line 650 for track changes, line 622 for final). 

2. Lines 130-7 of introduction seems like methods to me. If you could take this part then the paragraph you be as follows, that way you would say what and why you did on your research.

“To evaluate the effectiveness of different summary types for people with different careers, we created a survey that presents participants with a video abstract, graphical abstract, plain language summary, or published abstract from two papers in the same subject area. The survey looked at comprehension, perceived understanding, enjoyment, and whether the participants wanted to see more summaries of that type (…)”

We took this suggestion and removed the more methods-like components of the introduction. (Line 130)

3. Line 238: “input their gender if they so desired (a fill-in-the-blank that was not required)”

Were there many people who didn’t fill this? Once this was something that you reported I think this topic should have been mandatory, but that is not something you can change at this point. I Believe, then, that this (the number of people that didn’t answer this topic) should be reported when you inform the proportion of men/women that answered the survey.

505 of 538 reported a binary gender which was then used for the proportion of men/women who answered the survey. Data from individuals that either didn’t report a gender or reported a non-binary gender were still used for analysis, but were left out of gender-specific reporting. 

The proportion of participants reporting a binary gender is now included in the manuscript at (Line 323 in track changes, line 314 in final). 

4. Line 367: “Given the clear reported preference for written forms of communication (fig 2A), it was expected that the plain language summaries and perhaps the published abstracts would be the most effective summaries when tested (…) but videos were either as effective or more effective than plain language summaries in all cases except comprehension of science related participants for the Cohn et al. paper where plain language summaries had a higher average score (M=4 of 6 for video, M=5 of 6 for plain language) (fig 3)”

I believe this deserves some kind of discussion. Although the most preferable format is the written one, the videos were the easier format for comprehension. How would the authors explain that? A recently published paper from physiotherapy journal (https://doi.org/10.1016/j.physio.2018.11.003) analysed all the plain language summaries from the Physiotherapy Evidence Database and it found that only 2% of reports were considered at a suitable reading level for non scientific population! That could explain why the scientific community in addition to prefer, comprehend the message better than the rest of the people surveyed.

It is interesting that the most preferred format was written, but the videos were best for comprehension. We feel that written summaries were reported as the most preferred because they require the least effort and because written summaries can be skimmed. We don’t have direct evidence for this from the survey, but the comments on the survey and in talks with other scientists this definitely seems to be the difference. Also, many of the commenters on the video surveys said that they didn’t know just how helpful a video abstract could be which perhaps suggests that they aren’t widely watched. 

Overall, there is still a lot of research to be done on this subject and we hope this paper opens people’s eyes to how much we need data. 

To specifically address this point, we are now reporting the Flesh Reading Scores for both plain language summaries and published abstracts as calculated in the same way as the paper you mentioned. (New Table 1, line 175 track changes, line 166 final) 

The plain language summary scores are 63 and 58 for Cohn et al and Takata et al respectively, which are both considered a normal level and suitable for 9/10th grade. This is a bit higher than the recommended 6th grade reading level to reach the general public, but the Flesch Reading criteria are often harsh on scientific writing where you often need to use unfamiliar words for protein names or other phenomena. We also put our plain language summaries through other calculators and got a variety of scores, so this may not be the final word on whether a plain language summary is good/effective or not. We have commented on this idea in the discussion as well (line 635 track changes, line 625 final). 

We are also providing suggestions for authors of plain language summaries which reference for the eLife and the Cochrane guidelines. Also in the manuscript, we reference a jargon detector that may be helpful in limiting unfamiliar words and increasing the readability of plain language summaries. (line 631 in track changes, line 621 in final).

5. I understand the authors say they found a significant result on the correlation analysis and for that found it important to report, however, as the authors themselves report on the text, the correlation could have happened because one of the papers is harder to read once it requires more background knowledge to understand. Therefore, although significant this should be interpreted carefully as this may not reflect the true situation, mainly because these sort of resources usually target people that are not part of the scientific community or people from other areas of research, which, as consequence, need an easier text to understand the results of a very specific scientific study. 

Also you could not analyse the correlation for plain language summaries, perhaps it would not be correlated since almost everyone preferred PLS but not all comprehend them. Again, if the plain language summaries were really easier to read (which they appear they are not) the preferred format could be the best for understanding (-but that is just a theory!)

To address this point, we have reported the Flesch Reading scores for both plain language summaries (New Table 1, line 175 track changes, line 166 final). These reading scores do not suggest that one of the plain language summaries was far more difficult to read than the other. We believe the difference between the two papers has to do with the amount of background knowledge needed to understand the implications of the findings. In general, any form of reading score doesn’t really measure how easy it is to get the point of an article. You can write an explanation of quantum mechanics using only the 1000 most common English words, short sentences, and active voice, but that doesn’t guarantee that people will understand quantum mechanics. They will be able to literally read what you wrote, but that’s not necessarily what is most important. We have added a small change to our discussion to be clear about this point (line 542 track changes, line 532 final).

That said, authors of plain language summaries should definitely try to keep their writing easy to read when possible and we have made sure that our manuscript reflects that opinion.

---

## [Editor Report · Decision Letter 2]

21 Oct 2019

Video abstracts and plain language summaries are more effective than graphical abstracts and published abstracts

PONE-D-19-16933R2

Dear Dr. Bredbenner,

We are pleased to inform you that your manuscript has been judged scientifically suitable for publication and will be formally accepted for publication once it complies with all outstanding technical requirements.

I am still waiting for an answer regarding the possibility of including a video abstract for the article.

With kind regards,

David Orrego-Carmona, Ph.D.

Academic Editor

PLOS ONE
---

## [Editor Report · Acceptance letter]

7 Nov 2019

PONE-D-19-16933R2 

Video abstracts and plain language summaries are more effective than graphical abstracts and published abstracts 

Dear Dr. Bredbenner:

I am pleased to inform you that your manuscript has been deemed suitable for publication in PLOS ONE. Congratulations! Your manuscript is now with our production department. 

With kind regards,

on behalf of

Dr. David Orrego-Carmona 

Academic Editor

PLOS ONE